# Temporal Variation of Soil Moisture and Its Influencing Factors in Karst Areas of Southwest China from 1982 to 2015

**Xiaocha Wei [1], Jixi Gao [2,*], Sihan Liu [2,*] and Qiuwen Zhou [1]**

[1] School of Geography and Environmental Science, Guizhou Normal University, Guiyang 550001, China; xiaocha_wei@163.com (X.W.); zqw@gznu.edu.cn (Q.Z.)

[2] Satellite Application Center for Ecology and Environment, Ministry of Ecology and Environment, Beijing 100094, China

* Correspondence: gjx@nies.org (J.G.); liusihan1200@163.com (S.L.); Tel.: +86-010-58311599 (J.G.); +86-010-58311528 (S.L.)

**Abstract:** Climate change and human activities are two dominating factors affecting soil moisture temporal variation trends, whereas their individual contributions to soil moisture trends still remain uncertain in the karst areas of Southwest China. Based on the linear regression trend analysis, Mann–Kendall mutation detection, and residual analysis methods, we quantified the contributions of climate change and human activities to soil moisture temporal variation trends in the karst areas of Southwest China. The results showed that the soil moisture in the study area experienced a drying trend from 1982 to 2015. The mutated year was 1999, and the soil moisture decreasing trend was more evident from 2000 to 2015 than from 1982 to 1999. Human activities and climate change accounted for 59% and 41%, respectively, of soil moisture drying trends. Owing to the spatial heterogeneity of geomorphic features, the individual contributions of climate change and human activities to soil moisture trends exhibited regional differences. Although remarkable regional vegetation restoration was found since applying the Grain for Green Project, the negative impact of vegetation restoration on soil moisture cannot be neglected. This study is a quantitative analysis of the relative impacts of climate change and human activities on soil moisture trends, and our findings provide a theoretical reference for the sustainable use of soil water resources in the karst areas of Southwest China.

**Keywords:** soil moisture; climate change; human activities; karst area





## 1. Introduction

Soil moisture is a critical variable for Earth's systems and plays an important role in local ecological environments, especially in the karst areas [1]. Owing to the unique hydrogeographic features, the karst areas of Southwest China are characterized by a thin soil layer and high water penetration [2,3], and soil moisture is the dominant restricting factor [4,5]. In recent years, increasing impacts of human activities and global climate change have significantly influenced soil moisture dynamics, which may increase the risk of water resource storage and degradation and other eco-environmental problems [6–8]. Therefore, studying the soil moisture changes and its influencing factors can be significant for regional water resource management and ecological restoration in the karst areas of Southwest China.

Many scholars have studied the temporal variation trends of soil moisture and its influencing factors. For example, based on the "space series instead of time series" method, Zhou et al. highlighted that the succession of grassland to shrub and woodland would decrease the soil moisture content, and the shrubland is characterized by a more robust soil water conservation capacity than other land types in dry seasons [9,10]. Fu et al. found that the soil moisture content of natural forestland was higher than that of abandoned and sloping farmlands, and the artificial forestland exhibited the lowest soil moisture content [11]. Zhang et al. found that precipitation, soil evaporation, and vegetation

transpiration affected the changes in soil moisture in the karst peak cluster depression area, and the effect of precipitation changes on soil moisture fluctuations is the most evident [12].

In general, previous studies have confirmed that climate factors and human activities significantly influence the temporal variation in soil moisture [13–16]. However, these studies were primarily conducted on a plot and slope scale, and few studies focused on the regional scale. The karst areas of Southwest China are typically characterized by rugged terrain and strong land surface spatial heterogeneity, resulting in significant variability in soil moisture [17–19]. Therefore, revealing these plot and slope scale results on a regional scale is challenging.

On a regional scale, based on the linear regression trend analysis method, Deng et al. found that because of warming and vegetation recovery, the soil moisture in the karst areas of Southwest China showed a decreasing trend from 1979 to 2017. However, they did not quantify the impact of climate change and vegetation restoration on soil moisture drying. Additionally, Wei et al. highlighted that the remarkable vegetation restoration was the primary reason for the soil moisture drying the karst areas of Southwest China [8]. However, they evaluated the effects of vegetation restoration on regional soil moisture content dynamics in paired years with similar precipitation conditions but did not quantify the individual impacts of climate change and human activities on soil moisture variation. According to previous studies, the contributions of climate change and human activities to soil moisture temporal variation trends remain uncertain in the karst areas of Southwest China.

The karst areas of Southwest China experience a subtropical, humid monsoon climate, and climate change significantly affects the soil moisture [20–24]. This region is typically characterized by a thin soil layer, severe rocky desertification, and fragile ecological environment [25–27]. China's state and local governments have launched ecological restoration projects to protect the local ecological conditions, including the Natural Forest Protection Project, the Grain for Green Project, and the Karst Rocky Desertification Comprehensive Control and Restoration Project. In recent decades, remarkable regional vegetation restoration has occurred in this area [28–31] and the vegetation recovery will continue in the future, which may significantly change the local hydrological processes, such as evapotranspiration and rainfall interception, and directly or indirectly affect soil moisture content. Thus, under global climate change, vegetation restoration, and more water resource consumption, quantifying the influence of climate change and human activities on regional soil moisture content in the karst areas of Southwest China is critical for ecological restoration and water management.

Therefore, this study quantifies the individual contributions of climate change and human activities on temporal variation trends of soil moisture in the karst areas of Southwest China. First, the temporal variation trend of soil moisture was evaluated using the linear regression trend analysis and Mann–Kendall mutation detection methods. Second, the contributions of climate change and human activities were quantified using the residual analysis method. This study could support water resource management and accelerated human activities in a changing climate in the karst areas of Southwest China.

## 2. Materials and Methods

### 2.1. Study Area

The study area is in Southwest China and covers approximately 800,000 km$^2$ of the total land surface (97.5° E–112° E, 21.1° N–29.2° N), including the Yunnan and Guizhou Provinces and the Guangxi Zhuang Autonomous Region, China. The study area is distributed with complex terrain, and the elevation decreases from northwest to southeast (Figure 1). According to previous studies, this study area can be divided into eight landform types: peak forest plain, karst gorge, non-karst region, karst basin, middle-high hill, peak cluster depression, karst trough valley, and karst plateau [32]. Red, yellow, yellow–brown, brown, and lime soils dominate the soil types. The vegetation types primarily contain subtropical and tropical mountain coniferous forests, evergreen and deciduous

broad-leaved mixed forests, and shrubs. The annual mean precipitation is 1332.63 mm, and the annual mean temperature is 17.14 °C. Evident differences occur in precipitation and air temperature distributions, characterized by decreasing from the southeast and southwest to the middle and northwest of this region.

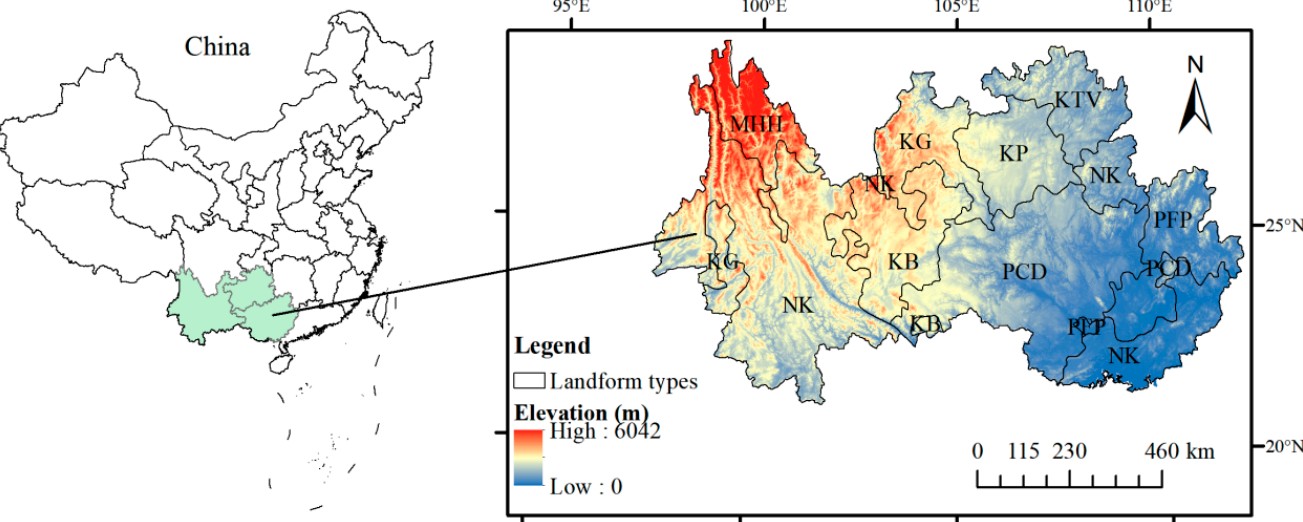

**Figure 1.** The location and elevation in the karst areas of Southwest China (the administrative map was obtained from the National Earth System Science Data Center (http://www.geodata.cn/, accessed on 15 August 2018)). The elevation data were obtained from Geospatial Data Cloud (http://www.gscloud.cn/search, accessed on 9 March 2021). The landform type data were obtained from the previous study by Wang et al. [32]) (PFP: peak forest plain; KG: karst gorge; NK: non-karst region; KB: karst basin; MHB: middle-high hill; PCD: peak cluster depression; KTV: karst trough valley; KP: karst plateau).

### 2.2. Data and Processing

#### 2.2.1. Soil Moisture

The soil moisture content was derived from the ERA-Interim dataset, derived from the European Centre for Medium-Range Weather Forecasts (ECMWF) [33–35]. This dataset was used in many studies conducted in the Tibet Plateau, Loess Plateau, and the karst areas of Southwest China, and it well matched the filed measured soil moisture [36–38]. The ERA-Interim dataset contains the soil moisture content data of four soil layers (0–7 cm, 7–28 cm, 28–100 cm, and 100–189 cm). The surface soil moisture (0–7 cm) from 1982 to 2015 was used in this study, the metric is $m^3/m^3$, and this dataset was resampled as the Albers Equal-area Conic projection and WGS1984 ellipsoid, with a spatial resolution of 10 km.

#### 2.2.2. Precipitation and Temperature

The precipitation and temperature data were downloaded from the China Meteorological Science Data Sharing Service Network (http://data.cma.cn/, accessed on 18 March 2021). The spatial precipitation and temperature data were the thin-plate spline spatial interpolation method using the Australian National University Spline software. Then they were resampled as the Albers Equal-area Conic projection and WGS1984 ellipsoid, and with a spatial resolution of 10 km.

#### 2.2.3. The Normalized Difference Vegetation Index (NDVI)

The NDVI dataset was derived from advanced very-high-resolution radiometer (AVHRR) global inventory modeling and mapping studies (GIMMS) NDVI3g.v1 (https://ecocast.arc.nasa.gov/data/pub/gimms/3g.v1/, accessed on 4 October 2021), with a spatial resolution of 8 × 8 km [39], and the spatial resolution was resampled to 10 km. This

is the most extended global vegetation dataset and is widely used in regional vegetation change research [40–42].

### 2.2.4. Land Use and Land Cover

This study's land use and land cover data include data from the years of 1980, 2000, and 2015, obtained from the Resource and Environment Science and Data Center (http://www.resdc.cn/, accessed on 18 October 2021), with a spatial resolution of 1 km. This dataset was resampled as the Albers Equal-area Conic projection and WGS1984 ellipsoid, with a spatial resolution of 10 km.

### 2.3. Methods

#### 2.3.1. Linear Regression Trend Analysis Method

Based on the linear regression analysis method, the interactive data language program evaluated the temporal variation trend of the soil moisture, precipitation, temperature, and NDVI by judging the regression equation slope [43]. The specific algorithms are

$$\beta = \text{Median}\left(\frac{x_i - x_j}{i - j}\right) \tag{1}$$

In this equation, $1 < j < i < n$, Median represents the median, and $\beta$ represents the regression equation slope. If $\beta > 0$, the soil moisture experienced an increasing trend, if $\beta < 0$, the soil moisture experienced a decreasing trend, and if $\beta = 0$, the soil moisture experienced a stable trend.

#### 2.3.2. Mann–Kendall Mutation Detection Method

Mann–Kendall is a mutation detection method with a wide detection range, less artificiality, and a high quantization degree. It does not require the samples to follow a specific distribution, nor is it disturbed by a few outliers [43]. The mutated year of the temporal variation trends of soil moisture, precipitation, temperature, and NDVI was detected using MATLAB based on the Mann–Kendall method.

#### 2.3.3. Residual Analysis Method

The residual analysis method can quantify the individual contributions of climate change and human activities to soil moisture trends [43]. First, the entire study period was divided into a reference period and a conservation period based on the mutated year judged by using the Mann–Kendall method. Then, the soil moisture, precipitation, and temperature were used to build a linear regression formula in the reference period in Equation (2) as follows. Based on this formula, the precipitation and temperature in the conservation period were input into the formula to simulate the soil moisture. The influence of human activities on soil moisture changes was judged according to the difference between observed soil moisture in the reference period and the simulated soil moisture in the conservation period. The contribution of human activities was the slope of residual soil moisture divided by the observed soil moisture. The contribution of climate change was the difference of 100% and the contribution of human activities. The specific algorithms of the residual analysis method are as follows:

$$SM_{(i,t)} = aT_{(i,t)} + bP_{(i,t)} + c \tag{2}$$

$$SM_{\text{residual}} = SM_{\text{observed}} - SM_{\text{simulated}} \tag{3}$$

$$C_{h(i,t)} = \frac{\text{Slope}\,(SM_{\text{residual}})}{\text{Slope}\,(SM_{\text{observed}})} \tag{4}$$

$$C_{c(i,t)} = 1 - C_{h(i,t)} \tag{5}$$

In this formula, $SM_{(i,t)}$ represents the soil moisture in t year in i pixel, $a$ represents the regression coefficient of $SM_{(i,t)}$ at temperature $T_{(i,t)}$, $b$ represents the regression coefficient of

$SM_{(i,t)}$ at precipitation $P_{(i,t)}$, and $c$ is a constant value. $SM_{residual}$ is the residual soil moisture, which is the difference between the observed and simulated soil moisture. $SM_{residual} > 0$ and $SM_{residual} < 0$ represent the positive and negative impacts of human activities on soil moisture trends, respectively. Slope is the slope of the data series, $C_{h(i,t)}$ is the contribution of human activities to soil moisture change, and $C_{c(i,t)}$ is the contribution of climate change to soil moisture trends.

## 3. Results

### 3.1. Temporal Variation Characteristics of Soil Moisture Content

As shown in Figure 2a, the annual mean soil moisture in the karst areas of Southwest China from 1982 to 2015 experienced a drying trend with a change rate of $-0.004$ $(m^3/m^3)/$ 10a ($R^2 = 0.47$). The annual mean soil moisture content was 0.237 $m^3/m^3$, with the highest soil moisture content of 0.248 $m^3/m^3$ in 1983 and the lowest of 0.223 $m^3/m^3$ in 2009. The soil moisture temporal variation exhibited two prominent stages, among which the change rate of soil moisture trends was $-0.003$ $(m^3/m^3)/10a$ ($R^2 = 0.16$) from 1982 to 1999 and $-0.004$ $(m^3/m^3)/10a$ ($R^2 = 0.119$) from 2000 to 2015. The soil moisture drought in 2000–2015 was more evident than in 1982–1999. Furthermore, Figure 2b shows that 1999 was the abrupt year of soil moisture change in the karst areas of Southwest China. Figure 2a,b indicates that the soil moisture in the karst areas of Southwest China experienced a drying trend from 1982 to 2015, and that the soil moisture drought was particularly significant since 1999.

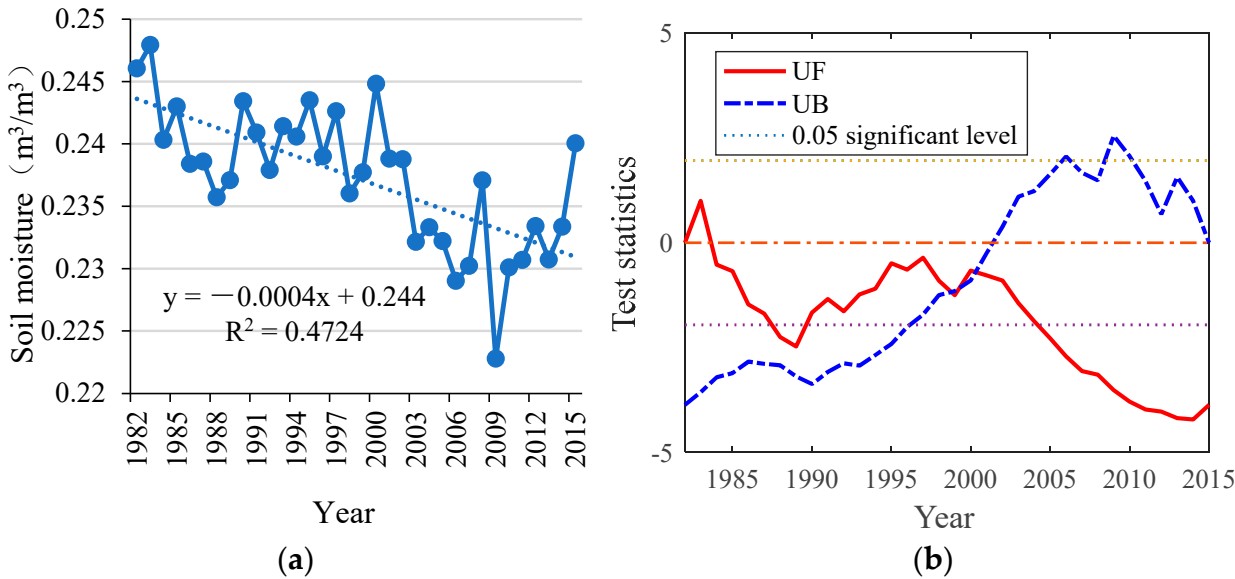

**Figure 2.** Temporal variation (**a**) and mutation detection (**b**) of the annual mean soil moisture in the karst areas of Southwest China from 1982 to 2015.

### 3.2. Temporal Variation Characteristics of Precipitation and Temperature

The precipitation in the karst areas of Southwest China from 1982 to 2015 exhibited a fluctuating downward trend with a change rate of $-22.85$ mm/10a (Figure 3a). The annual mean precipitation was 1332.63 mm, with the highest precipitation recorded in 1994 (1534.25 mm) and the lowest recorded in 2011 (1067.23 mm). The temperature showed a significant upward trend from 1982 to 2015 of 0.30 °C/10a (Figure 4a). During the study period, the average annual temperature was 17.14 °C, the highest temperature was recorded in 2009 at 17.88 °C, and the lowest was recorded in 1984 at 16.38 °C. Overall, the climate in the karst areas of Southwest China showed a warming and drying trend from 1982 to 2015. Figure 3b shows that there are multiple intersection points of annual precipitation in the karst areas of Southwest China during 1982–2015, and they are 1986, 1990, and 2003, respectively. It indicates that the change in the annual mean precipitation is unstable.

Figure 4b shows there is one mutation year of the annual mean temperature trend in 1997, and the temperature presented a more stable variation trend than the precipitation.

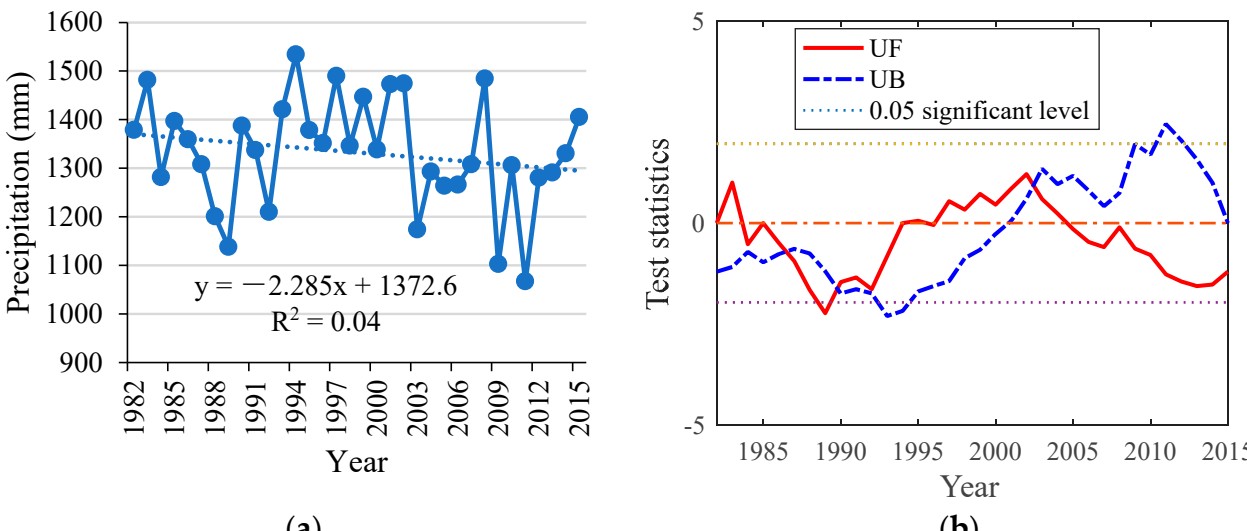

**Figure 3.** The temporal variation (**a**) and mutation detection (**b**) of the annual mean precipitation in the karst areas of Southwest China from 1982 to 2015.

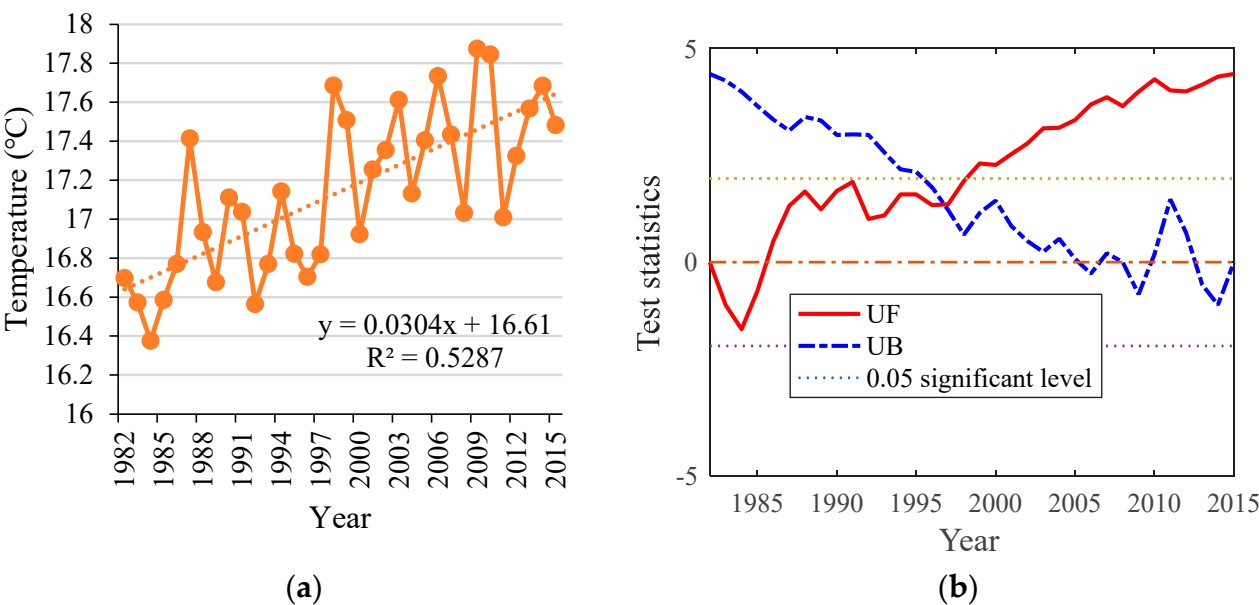

**Figure 4.** Temporal variation (**a**) and mutation detection (**b**) of the annual mean temperature in the karst areas of Southwest China from 1982 to 2015.

### 3.3. Temporal Variation Characteristics of NDVI

The annual mean NDVI in the karst areas of Southwest China from 1982 to 2015 showed a fluctuating upward trend with a change rate of 0.008/10a (Figure 5a). The multiyear average NDVI in the study area was 0.65, with the highest NDVI in 2015 (0.67) and the lowest in 1984 (0.63). Figure 5b shows the mutation detection of the annual mean NDVI, and the mutation year was 1999. The results indicate that the prominent increases in vegetation cover occurred in the karst areas of Southwest China from 1982 to 2015, and there was a remarkable regional vegetation restoration effectiveness in the study areas since 1999.

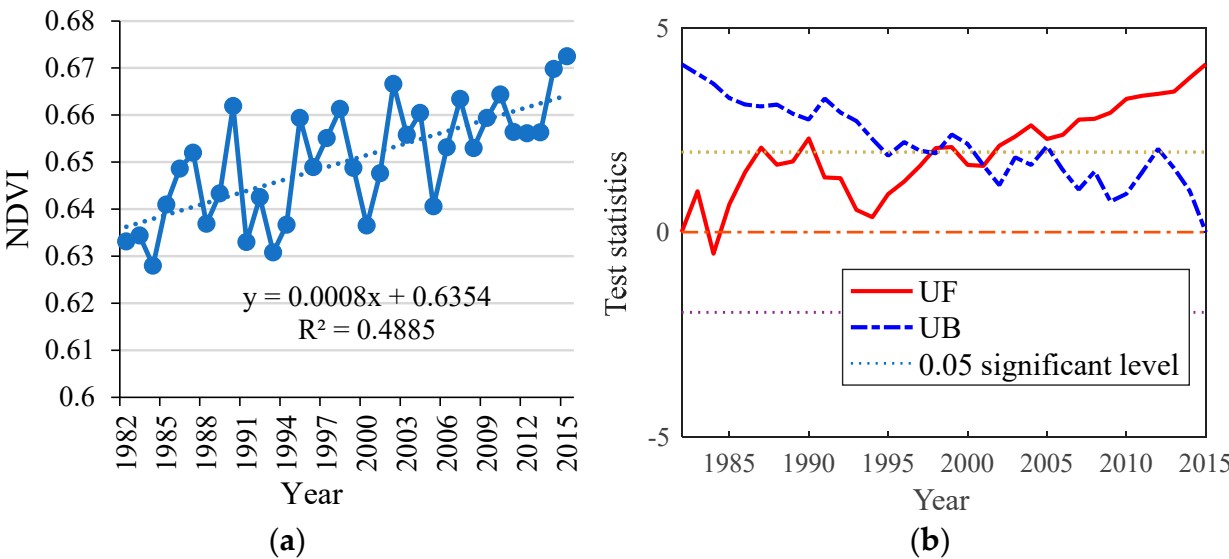

(**a**)                    (**b**)

**Figure 5.** Temporal variation (**a**) and mutation detection (**b**) of the annual mean NDVI in the karst areas of Southwest China from 1982 to 2015.

*3.4. The Contributions of Climate Change and Human Activities to Soil Moisture Trends*

The mutation test results of soil moisture temporal variation trends in Section 3.1 showed that the mutation year is 1999, and China's state and governments started implementing vegetation restoration projects in the karst areas of Southwest China in 1999. Therefore, this manuscript takes 1999 as the change point for modeling a linear regression equation of climate and soil moisture in the reference period (1982–1999), and based on this climate–soil moisture regression model, the soil moisture in a conservation period (2000–2015) was simulated. Figure 6 shows that the observed soil moisture primarily decreased from east and southwest to northwest, and the low soil moisture content was mainly distributed in the middle-high hill areas in the northwest of the study area. Figure 7 shows the spatial distribution of simulated soil moisture, and the high soil moisture content was primarily distributed in the southeast and southwest, and low soil moisture in the central and northwest of the study area. In general, the spatial distribution of observed soil moisture was consistent with the simulated soil moisture.

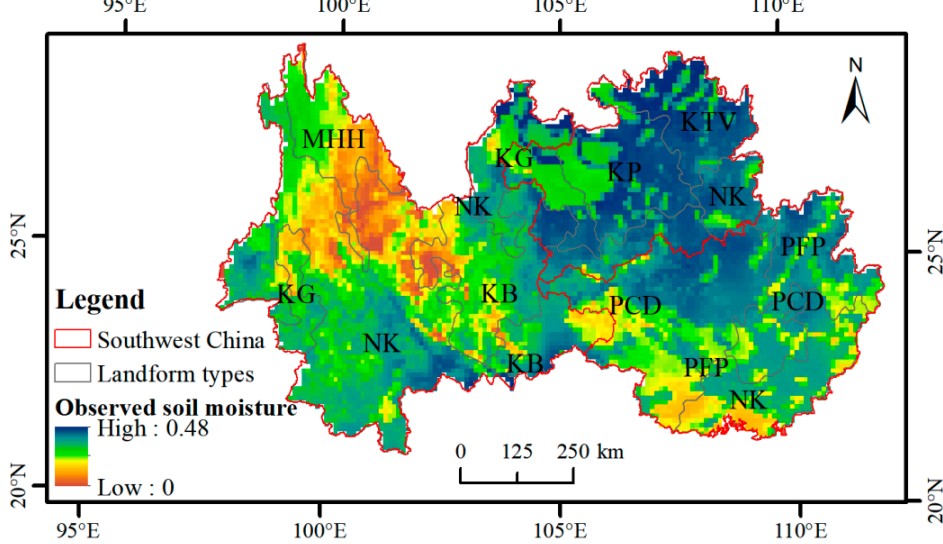

**Figure 6.** Observed soil moisture in the karst areas of Southwest China from 2000 to 2015 (PFP: peak forest plain; KG: karst gorge; NK: non-karst region; KB: karst basin; MHB: middle-high hill; PCD: peak cluster depression; KTV: karst trough valley; KP: karst plateau).

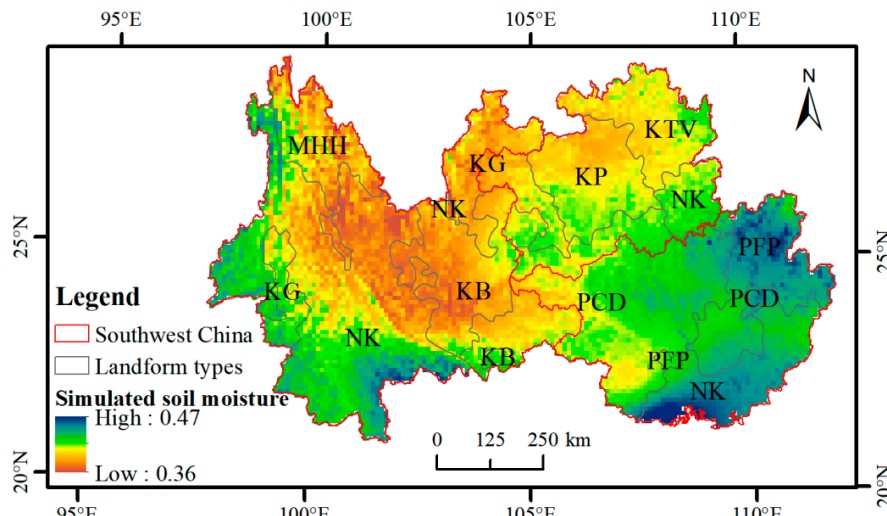

**Figure 7.** Simulated soil moisture in karst areas of Southwest China from 2000 to 2015 (PFP: peak forest plain; KG: karst gorge; NK: non-karst region; KB: karst basin; MHB: middle-high hill; PCD: peak cluster depression; KTV: karst trough valley; KP: karst plateau).

Figure 8 shows that the residual soil moisture in the study area from 2000 to 2015 was overall lower than 0, indicating that human activities negatively affected the soil moisture. The negative impact of human activities on soil moisture content was primarily distributed in the middle-high hill in the northern Yunnan Province and the non-karst region in southern Guangxi. The residual soil moisture in the landform types decreased in order as follows: karst gorge > karst plateau > karst trough valley > peak cluster depression > karst basin > peak forest plain > non-karst region > middle-high hill (Figure 9). Figure 9 shows that the average contribution of human activities to soil moisture change from 2000 to 2015 was 59%, and the average contribution of climate change to soil moisture change was 41%. The results indicate that human activities primarily affected the soil moisture drying trend in the karst areas of Southwest China from 2000 to 2015. In the different landform types, the contribution of human activities to the soil moisture trend was primarily distributed in the middle-high hill, non-karst region, karst trough valley, karst basin, and peak cluster depression. However, the contribution of climate change was primarily distributed in the peak forest plain, karst plateau, and karst gorge (Figure 9).

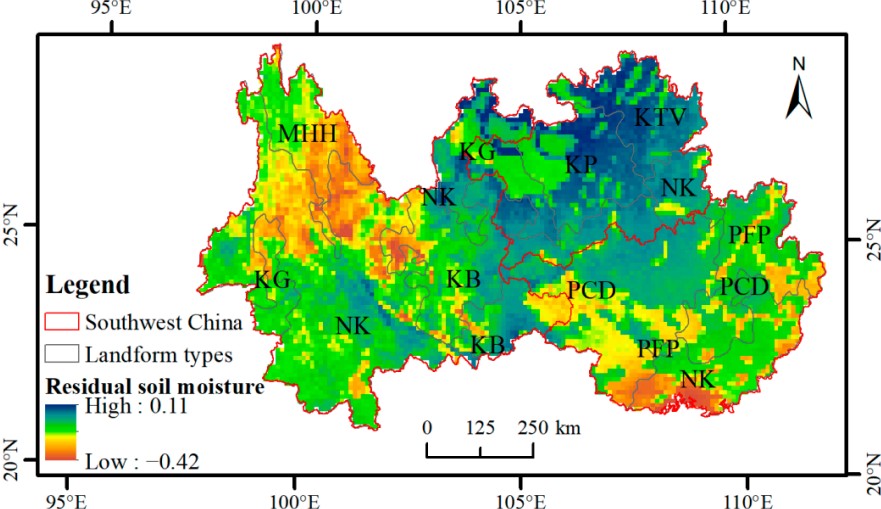

**Figure 8.** Residual soil moisture in the karst areas of Southwest China from 2000 to 2015 (PFP: peak forest plain; KG: karst gorge; NK: non-karst region; KB: karst basin; MHB: middle-high hill; PCD: peak cluster depression; KTV: karst trough valley; KP: karst plateau).

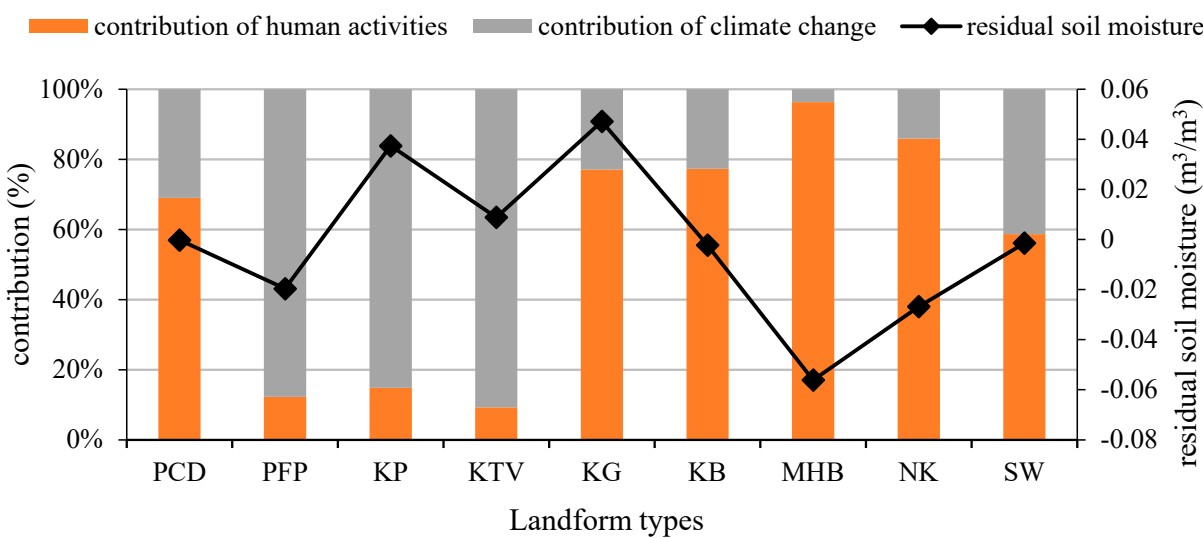

**Figure 9.** Residual soil moisture and the contributions of human activities and climate change to soil moisture trends in the karst areas of Southwest China from 2000 to 2015 (PFP: peak forest plain; KG: karst gorge; NK: non-karst region; KB: karst basin; MHB: middle-high hill; PCD: peak cluster depression; KTV: karst trough valley; KP: karst plateau).

## 4. Discussion

### 4.1. The Difference in Contributions of Climate Change and Human Activities to Soil Moisture Trends

The results in this study indicate that the soil moisture in the karst areas of Southwest China experienced a drying trend from 1982 to 2015, especially since 1999. The residual value of soil moisture in the karst areas of Southwest China was lower than 0, and the contribution of human activities and climate change to soil moisture trends was 59% and 41%, respectively. Moreover, the contribution varies among different landform types. Human activities primarily affected the soil moisture trends in high vegetation cover areas, and climate change affected the low vegetation cover areas. Human activities have negatively affected the soil moisture in the karst areas of Southwest China, particularly since 1999. The reasons might be as follows.

First, as shown in Figures 10 and 11, the construction land areas in the karst areas of Southwest China increased from 1980 to 2015; the increased areas were pronounced from 2000 to 2015. With the acceleration of urbanization and the increase in population, the demand for water resources, such as living, irrigation, and groundwater overexploitation, has increased, decreasing the soil moisture. Simultaneously, owing to the increase in the impermeable land surface, such as construction land, the precipitation formed a high surface runoff, and low precipitation infiltrated the ground, eventually reducing the soil moisture.

Second, from 2000 to 2015, the forestland in the karst areas of Southwest China showed an increasing trend (Figures 10 and 11), and the NDVI primarily exhibited an increasing trend (Figure 5a), negatively correlating with the trend of soil moisture content. The results demonstrated that vegetation restoration might also be vital for soil moisture drying in the region. The reason might be that vegetation restoration will lead to more water resource consumption through stronger rainfall interception, vegetation transpiration, and soil evapotranspiration loss, reducing soil moisture content. Many studies have shown that under global vegetation restoration [44], the excessive consumption of soil moisture by large-scale vegetation restoration will cause severe soil moisture deficits. Zhou et al. [2] highlighted that root growth increases the soil's infiltration capacity after the forestland restoration, and the increase in plant density and trunk diameter causes increased water consumption. Furthermore, the Loess Plateau and other areas have shown new forest–water contradictions, such as 'soil dry layer', 'little old man tree', and land degradation, which have an unsustainable impact on the surface ecosystems [45,46]. Some scholars have found that in small watersheds, river basins, and regional spatial scales, the average

vegetation afforestation range reduces the land surface runoff by 50–60%, and with an increase in the drought degree, it can reach 100%. Runoff occurs due to the consumption of water sources other than precipitation by vegetation restoration [47].

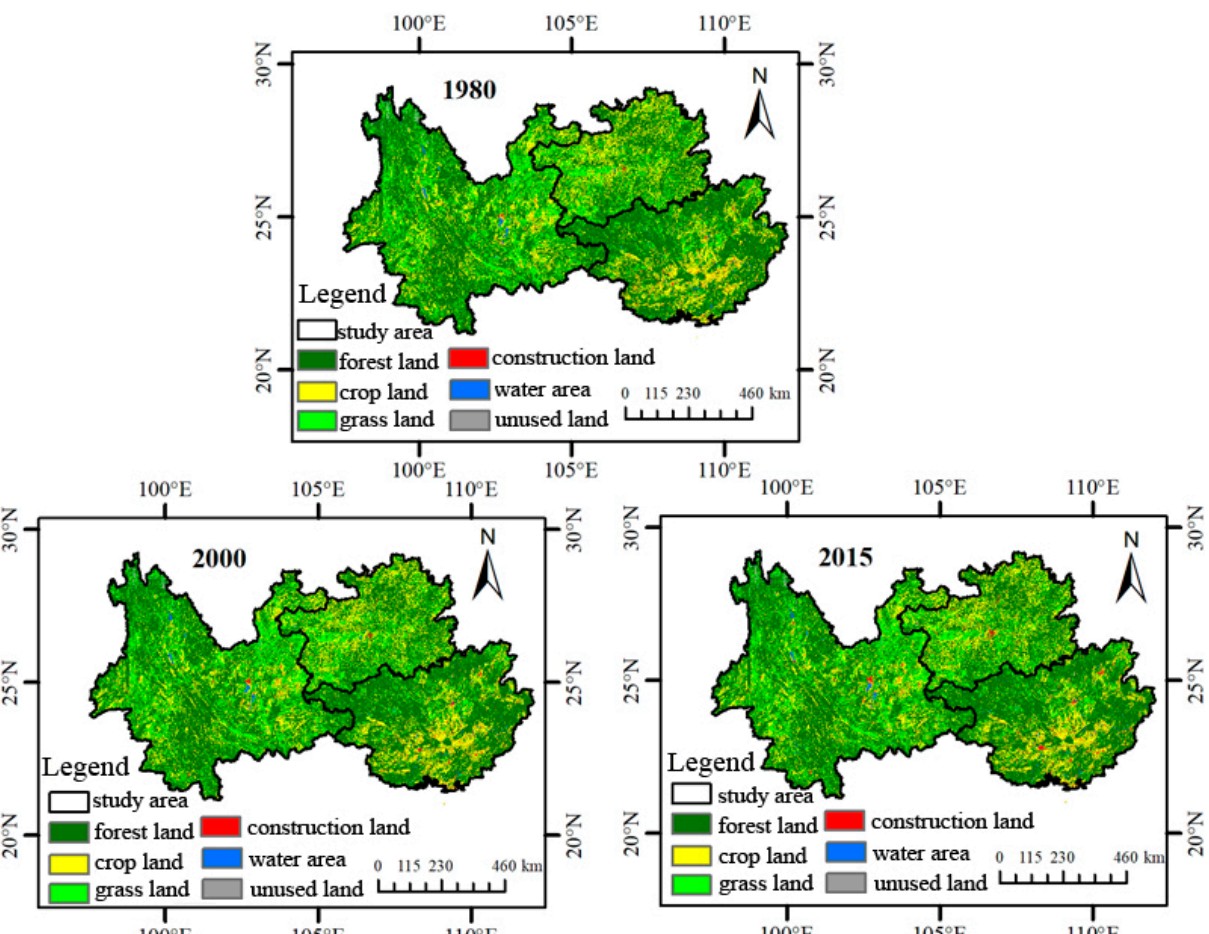

**Figure 10.** Spatial distribution of land use and cover change in the karst areas of Southwest China in 1980, 2000, and 2015.

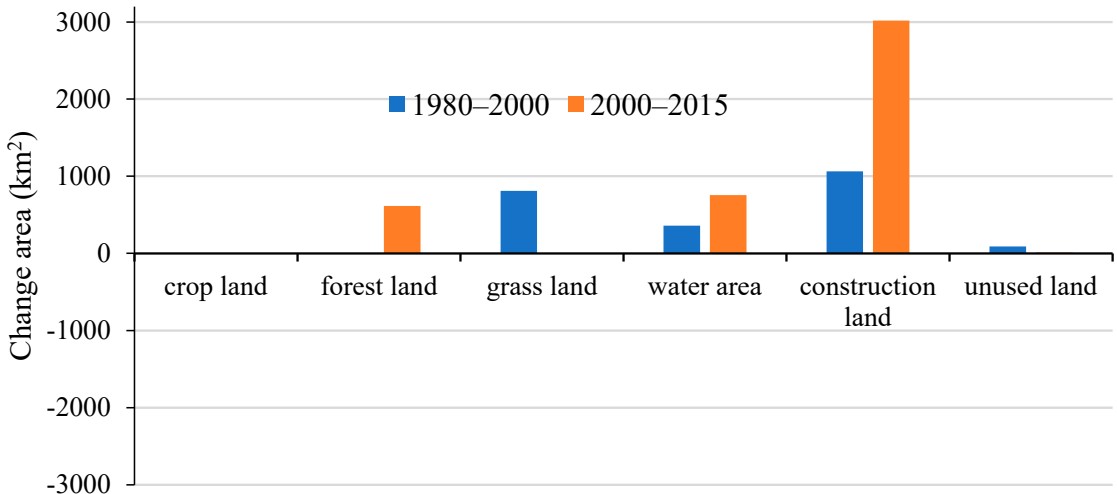

**Figure 11.** Land use and cover change in the karst areas of Southwest China from 1980 to 2000 and 2000 to 2015.

Furthermore, the contributions of climate change and human activities to soil moisture trends in the karst areas of Southwest China vary in different landform types. The contribution of human activities to soil moisture trends was primarily distributed in the middle-high hill, non-karst region, karst trough valley, karst basin, and peak cluster depression. The contribution of climate change to the soil moisture trend was primarily distributed in the peak forest plain, karst plateau, and karst gorge (Figure 9). Human activities primarily affected the soil moisture in high vegetation cover areas, and climate change affected the low vegetation cover areas. The reason might be that the vegetation cover was low in the middle-high hill, karst trough valley, karst basin, and peak cluster depression. After implementing ecological restoration measures, such as rocky desertification control projects, remarkable vegetation restoration occurred in those areas, negatively affecting soil moisture content. The vegetation coverage in non-karst areas is also high, negatively correlating with soil moisture storage. Several factors could be attributed to soil moisture trends, which can be summarized as effects of natural and human activities. The former is characterized by climate change, whereas the latter is represented by vegetation change. Therefore, the effect of human activities on soil moisture changes in these areas is more obvious due to vegetation changes, and the impact of climate change is not obvious. The vegetation cover is low in the peak forest plain, karst plateau, and karst gorge, where the climate change rather than human activities dominantly affected the soil moisture [48–54].

Due to the interdependent characteristics of climate change and human activities and their complex interaction, the dominant factors causing soil moisture drought still remain unclear in the karst areas of Southwest China. The present study separated the contributions of climate change and human activities on soil moisture trends, and the results indicate that human activities dominate the soil moisture trends, which is consistent with the research conducted by Feng et al. [17]. They pointed that land cover changes, especially the vegetation cover change, significantly influence regional soil moisture changes, while climate change dominated the global soil moisture. The contributions of climate change and human activities to soil moisture trends varied in different climatic zones, and their individual contributions strongly correlated to the local environmental characteristics and scale size. In the present study, we prove that the land cover changes such as vegetation restoration lead to regional soil moisture drought, while climate change plays limited influence on regional soil moisture trends. Our findings provide a theoretical reference for the sustainable use of soil water resources and climatic adaption in the karst areas of Southwest China on a regional scale. According to previous studies, the vegetation restoration in the karst areas of Southwest China is one of the most remarkable examples afforestation across the global scale [28–31], which will keep aggravating soil moisture drought and may increase the risk of water resource storage, land degradation, and other problems in the future. Therefore, the regulation of land use should be given attention to promote regional water management in the karst areas Southwest China. For example, an appropriate forest planning strategy is helpful for effective management of soil moisture storage and climatic adaption in this region.

*4.2. Limitations and Uncertainty*

Wang et al. highlighted that slope, slope position, and land use type are the main factors affecting soil moisture changes at the slope and small watershed scales [48]. Evapotranspiration, precipitation, and land use type are the primary factors influencing soil moisture changes at a regional scale. As the research scale increases, the dominant influences on soil moisture temporal variation trends by topographic factors gradually turn to climate factors [55]. Overall, the dominant factors affecting soil moisture trends differ at different scales [56,57]. However, this study analyzed only the soil moisture trends and their influencing factors in the karst areas of Southwest China on a regional scale. Therefore, the impacts of climate change and human activities on soil moisture trends have not been revealed on small scales, such as slope and watershed scales, in the karst areas of Southwest China. Moreover, soil moisture changes and their influencing factors will also differ in

different climatic zones [58,59], and this study only investigated the influence of climate change and human activities on soil moisture in the karst areas of Southwest China in the humid subtropical zone. Thus, the impacts of climate change and human activities on soil moisture trends should be further studied in karst areas in other climatic zones. Furthermore, this study only assessed the influencing factors on soil moisture changes at the surface soil layer (0–7 cm) in the karst areas of Southwest China. However, the temporal variation trends and their influencing factors in different soil layers are different. Therefore, this study's results only illustrate the impacts of climate change and human activities on surface soil moisture trends in subtropical humid karst areas, and have not been revealed at subsurface or deep soil layers.

Simultaneously, this study's results show that human activities are the dominant factor for soil moisture drought in the karst areas of Southwest China, and the negative impact of vegetation restoration on soil moisture cannot be neglected. This study used only NDVI and forestland area to characterize vegetation change. However, vegetation change is a complex ecological process, comprising the dynamic changes in vertical vegetation stratification, canopy density, leaf area index, vegetation stand structures, and other features. Additionally, the local topography and climate conditions affect vegetation dynamics. Future human activities and climate change will significantly affect soil moisture dynamics. Therefore, the effects of future climate change, vegetation restoration, and land use and cover change on soil moisture trends in the karst areas of Southwest China should be revealed using ecohydrological models in future research. Simultaneously, the spatial resolution of the remote-sensing data used in this study is low, which will have a mixed-pixel phenomenon in the mountainous karst area characterized by complex terrain, decreasing the accuracy of soil moisture inversion in this area. Therefore, in future research, using high-resolution remote-sensing data or building a soil moisture–environmental factor (lithology, soil, climate, and other factors) regress model using field monitoring data, which can more accurately characterize the soil moisture dynamics in complex terrain areas, is necessary.

### 4.3. The Contributions and Implications of This Study

The karst areas of Southwest China are typically characterized by high spatial heterogeneity [60,61], the soil moisture is primarily regionally varied, and its influencing factors are complex and diverse [17–19]. Climate change and human activities are two dominating factors affecting soil moisture temporal variation trends, but most studies have focused on small scales [9–12], such as plot, slope, and small watershed scales, which are unsuitable for illustrating the soil moisture trends and their influencing factors regionally. Overall, the individual contributions of climate change and human activities to temporal variation trends of regional soil moisture remain uncertain [13–16]. We analyzed the soil moisture trends and their influencing factors in the karst areas of Southwest China. The contributions of human activities and climate change to soil moisture trends were quantitatively distinguished, an essential supplement to the knowledge gap in soil moisture dynamics and its response to influencing factors.

This study's results indicate that accelerated human activities have caused soil moisture drying trends, and vegetation restoration will increase ecological water consumption and reduce the soil water content. These results have two implications. On the one hand, in the future practice of ecological construction, the reasonable vegetation restoration density and vegetation types, such as shrubs and low-density forest planning, should be selected according to local climate, topography conditions, and the comprehensive vegetation carrying capacity of soil water resources. On the other hand, socioeconomic activity is also critical for controlling the reasonable human population to ensure regional ecological safety and water resource security in the karst areas of Southwest China.

## 5. Conclusions

This study investigated the temporal variation trends of soil moisture in the karst areas of Southwest China from 1982–2015 and quantified the individual contributions of human activities and climate change to soil moisture trends. The results demonstrated that the soil moisture exhibited a drying trend from 1982 to 2015, and the decreasing trend of soil moisture from 2000 to 2015 was more obvious than from 1982 to 1999. The contribution of human activities to soil moisture trends in the karst areas of Southwest China from 2000 to 2015 was 59%, while the contribution of climate change was 41%. It indicates that the human activities dominate the soil moisture trends, which is the main reason that the regional soil moisture drought was aggravated in the karst areas of Southwest China, and the negative impact of vegetation restoration on soil moisture restoration cannot be neglected. Due to the spatial heterogeneity of geomorphic features, the individual contributions of human activities and climate change varied in different geomorphological types. Therefore, the negative effects of human activities to aggravate soil moisture drought should particularly be paid more attention. In conclusion, this study analyzed regional soil moisture trends and separated the contributions of its influencing factors in the karst areas of Southwest China, and the results could support ecological construction and water management in karst areas.

**Author Contributions:** Data curation, X.W.; conceptualization, J.G. and Q.Z.; formal analysis, X.W.; funding acquisition, J.G.; investigation, S.L.; methodology, Q.Z.; resources, J.G.; software, X.W.; writing—original draft, X.W.; writing—review and editing, J.G. and Q.Z. All authors have read and agreed to the published version of the manuscript.

**Funding:** This research was funded by the Scientific and Technological Research Project of Guizhou Province (grant number Qiankehe Jichu (2019) 1433 and Qiankehe Pingtai Rencai [2017]5726), the Natural Science Foundation of China (41801293), and the Joint Fund of the Natural Science Foundation of China and the Karst Science Research Center of Guizhou Province (grant number U1812401).

**Institutional Review Board Statement:** Not applicable.

**Informed Consent Statement:** Not applicable.

**Data Availability Statement:** Not applicable.

**Acknowledgments:** Thanks to all contributions to the Special Issue, the time spent by each author, and the anonymous reviewers and editorial managers who have greatly contributed to the development of the articles in this Special Issue.

**Conflicts of Interest:** The authors declare no conflict of interest.

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
