# Peer review of "Temporal Variation of Soil Moisture and Its Influencing Factors in Karst Areas of Southwest China from 1982 to 2015"

_water, doi:10.3390/w14142185_

Round 1

Reviewer 1 Report

This paper touches upon an important issue within soil moisture all external influences. It shows the influence of the climate change and human activity contributions to soil moisture temporal variation trends in the study area.  

This paper shows an innovative method applies to image analysis. The framework and the theoretical scheme are not quite clear. The conceptual and analytical framework is still lacking. The structure of this paper is acceptable.

Before providing detailed comments to the specific sections, I have some general suggestions to strengthen the analytical consistency.

Overall comment

The authors need to reframe the discussion section ad conclusion.  . Conclusion section still to be reframed.  The method section still to be structured mainly focus on analytical way

Many figure need to be removed or merged. The result section in too focus on figure an less on analysis. 

line 11: contribution to what?

line 14: check the spelling

line 38: Zhou et al... Provide the year

line 44:  Zhang et al… provide the year

line 85: ??? Be humble

line 105: provide the source of this figure

line 132: precise the resolution of this other dataset

line 171: the way the method is used is not clear. Mainly, there is a lack conceptual and analytical framework that will support the analysis

line 225: the way the result is present is not accurate and consistent, author insist on the presentation of figure instead of  using figure to support the analysis

line 387: i really have a problem with this way of presenting the discussion section.  it need to focus on comparing your results to others research.

line 402: the conclusion can more extensive. the perspective aspect is not developed

Some English editing are needed.

All remarks and comments are in the manuscript.

Hope these comments are helpful to improve the manuscript for submission in Water.

Author Response

Reviewer: 1

Comments and Suggestions for Authors:

This paper touches upon an important issue within soil moisture all external influences. It shows the influence of the climate change and human activity contributions to soil moisture temporal variation trends in the study area.  

This paper shows an innovative method applies to image analysis. The framework and the theoretical scheme are not quite clear. The conceptual and analytical framework is still lacking. The structure of this paper is acceptable.

Before providing detailed comments to the specific sections, I have some general suggestions to strengthen the analytical consistency.

The authors need to reframe the discussion section ad conclusion. Conclusion section still to be reframed.  The method section still to be structured mainly focus on analytical way.

Many figure need to be removed or merged. The result section in too focus on figure an less on analysis.

Reply: Thank you very much for your suggestions on how to improve our manuscript. According to these comments, we mainly focused on revising the method section, result section, discussion section, and conclusion section. I have revised made the point-by-point corrections below, and all the modifications are highlighted in red in the revised manuscript. Thank you for your helpful comments to improve our manuscript again, and we hope that the revised manuscript will meet with the submission in Water.

  1. line 11: contribution to what?

Reply: I have changed this sentence to “Climate change and human activities are two dominating factors affecting soil moisture temporal variation trends, whereas their individual contributions to soil moisture trends remain uncertain in the karst areas of Southwest China. .”

  1. line 14: check the spelling.

Reply: I have changed the sentence in line 14 “we quantified the climate change and human activity contributions to soil moisture temporal variation trends in the karst areas of Southwest China.” to “we quantified the contributions of climate change and human activities to soil moisture temporal variation trends in the karst areas of Southwest China.”

  1. line 38: Zhou et al... Provide the year.

Reply: I am sorry that after checking the stylistic in Water template, we found it dose not need to provide the year.

  1. line 44:  Zhang et al… provide the year.

Reply: According the stylistic in Water template, we found it dose not need to provide the year.

  1. line 85: ??? Be humble.

Reply: I have changed the sentence of “This study should provide a vital scientific basis for the rational use of soil and water resources in Southwest China’s karst areas and guide the evaluation of global climate change and human activity impacts on the water cycle process.” to “This study quantitative analysis of relative impacts of climate change and human activities on soil moisture trends, and our findings provide a theoretical reference for the sustainable use of soil water resources in the karst areas of Southwest China.”.

  1. line 105: provide the source of this figure.

Reply: I have provided the source of data in figure 1 in line 108-111 as “(The administrative map was obtained from the National Earth System Science Data Center (http://www.geodata.cn/). The elevation data was obtained from Geospatial Data Cloud (http://www.gscloud.cn/search). The landform types data was obtained from the previous study by Wang et al. [32])”.

  1. line 132: precise the resolution of this other dataset.

Reply: I have unified added the resolution of other data set in the revised manuscript as “the spatial resolution is 10 km”.

  1. line 171: the way the method is used is not clear. Mainly, there is a lack conceptual and analytical framework that will support the analysis.

Reply: The description of the method was not clear enough to be understood in before manuscript, I have changed the description of the methods in section 2.3.3 in the revised manuscript as follows:

2.3.3. Residual analysis method

The residual analysis method can quantify the individual contributions of climate change and human activities to soil moisture trends [43]. First, the entire study period was divided into a reference period and a conservation period based on the mutated year judged by using the Mann–Kendall method. Then, the soil moisture, precipitation and temperature were used to built a linear regression formula in the reference period as formula (2) as follows. Based on this formula, the precipitation and temperature in the conservation period were input into the formula to simulate the soil moisture. The influence of human activities on soil moisture changes was judged according to the difference between observed soil moisture in the reference period and the simulated soil moisture in conservation period. The contribution of human activities was the slope of residual soil moisture division the observed soil moisture. The contribution of climate change was the difference of 100% and the contribution of human activities. The specific algorithms of the residual analysis method are as follows:

, (2)

     , (3)

   ,  (4)

. (5)

In this formula, SM(i, t) represents the soil moisture in t year in i pixel, a represents the regression coefficient of SM(i, t) at temperature T(i, t), b represents the regression coefficient of SM(i, t) at precipitation P(i, t), and c is a constant value. SMresidual is the residual soil moisture, which is the difference between the observed and simulated soil moisture. SMresidual > 0 and SMresidual < 0, respectively, represent the positive and negative impacts of human activities on soil moisture trends. Slope is the slope of the data series, Ch(i, t) is the contribution of human activities to soil moisture change, and Cc(i, t) is the contribution of climate change to soil moisture trends.

  1. line 225: the way the result is present is not accurate and consistent, author insist on the presentation of figure instead of  using figure to support the analysis.

Reply: In the before manuscript, results were not properly described. In the revised manuscript, I have focused on describing the results according to the figures instead of  using figure to support the analysis. For example, I have changed the sentence in section 3.3 “It indicates that since the launch of ecological restoration projects, including the Natural Forest Protection Project, the Grain for Green Project, and the Karst Rocky Desertification Comprehensive Control and Restoration Project, the vegetation coverage in Southwest China’s karst areas began to increase in 1999.” to “The results indicate that the prominent increases in vegetation cover to occur in the karst areas of Southwest China from 1982 to 2015, and there was a remarkable regional vegetation restoration effectiveness in the study areas since 1999.”.

  1. line 387: i really have a problem with this way of presenting the discussion section.it need to focus on comparing your results to others research.

Reply: In the revised manuscript, I have compared the results at the present study with others research as follows:

Due to the interdependent characteristics of climate change and human activities and their complex interaction, the dominant factors causing soil moisture drought are still remain unclear in the karst areas of Southwest China. The present study separated the contributions of climate change and human activities on soil moisture trends, and the results indicates that the human activities dominates the soil moisture trends, which is consistent with the research conducted by Feng et al [17]. They pointed that land cover changes especially the vegetation cover change significantly influence on regional soil moisture changes, while the climate change dominated the global soil moisture. The contributions of climate change and human activities to soil moisture trends varied in different climatic zones, and their individual contributions strongly correlated to the  local environmental characteristics and scale size. In the present study, we prove that the land cover changes such as vegetation restoration leading to regional soil moisture drought, while the climate change plays limited influence on regional soil moisture trends, our findings provide a theoretical reference for the sustainable use of soil water resources and climatic adaption in the karst areas of Southwest China on a regional scale. According to previous studies, the vegetation restoration in the karst areas of Southwest China is one of the most remarkable afforestation across the global scale [28-31], which will keep aggravating soil moisture drought and may increase the risk of water resource storage, land degradation, and other problems in the future. Therefore, the regulation land use should be paid attention to promote regional water management in the karst areas Southwest China. For example, the appropriate forest planning strategy is helpful for effective management of soil moisture storage and climatic adaption in this region.

  1. line 402: the conclusion can more extensive. the perspective aspect is not developed.

Reply: According to the comments, I have concluded the main conclusion rather than simply repeat the results of this study. On the other hand, I have supplemented the implication and contribution of this study in the conclusion section. The detailed revision sections are highlighted in red in the revised manuscript as follows:

  1. Conclusions

This study investigated the temporal variation trends of soil moisture in the karst areas of Southwest China during 1982-2015, and quantified the individual contributions of human activities and climate change to soil moisture trends. The results demonstrated that the soil moisture exhibited a drying trend from 1982 to 2015, and the decreasing trend of soil moisture from 2000 to 2015 was more obvious than from 1982 to 1999. The contribution of human activities to soil moisture trends in the karst areas of Southwest China from 2000 to 2015 was 59%, while the contribution of climate change was 41%. It indicates that the human activities dominates the soil moisture trends, which is the main reason to aggravate the regional soil moisture drought in the karst areas of Southwest China, and the negative impact of vegetation restoration on soil moisture restoration cannot be neglected. Due to the spatial heterogeneity of geomorphic features, the individual contributions of human activities and climate change varied in different geomorphological types. Therefore, the negative effects of human activities to aggravate soil moisture drought should particularly be paid more attention. In conclusion, this study analyzed regional soil moisture trends and separated the contributions of its influencing factors in the karst areas of Southwest China, the results could support ecological construction and water management in karst areas.

  1. Some English editing are needed.Hope these comments are helpful to improve the manuscript for submission in Water.

Reply: The language has been carefully edited with grammar, spelling, word usage, sentence structure and general readability by a native English speaker with expertise in technical English editing.

Reviewer 2 Report

This is a very interesting article devoted to identifying the causes of changes in soil moisture over the past few decades. Using the karst areas of Southwest China as an example, the authors linked changes in the moisture content in soils with changes in the amount of precipitation. It is concluded that the decrease in the amount of soil moisture by 58.68% is due to climatic reasons, and by 41.32% due to economic activity. Accuracy up to a hundredth of a percent, of course, does not reflect reality, and in the article the estimates must be given with an accuracy of up to a percentage: 59 and 41%. The article can be recommended for publication after minor stylistic editing.

Author Response

Reviewer: 2

Comments and Suggestions for Authors:

This is a very interesting article devoted to identifying the causes of changes in soil moisture over the past few decades. Using the karst areas of Southwest China as an example, the authors linked changes in the moisture content in soils with changes in the amount of precipitation. It is concluded that the decrease in the amount of soil moisture by 58.68% is due to climatic reasons, and by 41.32% due to economic activity. Accuracy up to a hundredth of a percent, of course, does not reflect reality, and in the article the estimates must be given with an accuracy of up to a percentage: 59 and 41%. The article can be recommended for publication after minor stylistic editing:

Reply: I am very pleased that you are satisfied with our study and thank you very much for your suggestions to improve our manuscript. First, according to your suggestions, I have changed the contributions of human activities and climate change with 58.68% and 41.32% to an accuracy of 59% and 41% in the revised manuscript. Second, the language expression has been carefully edited with grammar, spelling, word usage, sentence structure and general readability by someone with expertise in technical English editing. The revised content are highlighted in red in our revised manuscript. Finally, thank you for your favorable comments again, and we hope that the revised manuscript will meet with your approval.
